# Evolution of the Internal Structure of Short-Period Cr/V Multilayers with Different Vanadium Layers Thicknesses

**DOI:** 10.3390/ma12182936

**Published:** 2019-09-11

**Authors:** Runze Qi, Qiushi Huang, Jiani Fei, Igor V. Kozhevnikov, Yang Liu, Pin Li, Zhong Zhang, Zhanshan Wang

**Affiliations:** 1Key Laboratory of Advanced Micro-Structured Materials MOE, Institute of Precision Optical Engineering, School of Physics Science and Engineering, Tongji University, Shanghai 200092, China; 2Shubnikov Institute of Crystallography of Federal Scientific Research Centre “Crystallography and Photonics” of the Russian Academy of Sciences, Leninskiy pr. 59, Moscow 119333, Russia

**Keywords:** multilayer, ultrathin metal films, columnar growth, soft X-rays

## Abstract

Cr/V multilayer mirrors are suitable for applications in the “water window” spectral ranges. To study factors influencing the internal microstructure of Cr/V multilayers, multilayers with different vanadium layers thicknesses varying from 0.6 nm to 4.0 nm, and a fixed thickness (1.3 nm) of chromium layers, were fabricated and characterized with a set of experimental techniques. The average interface width characterizing a cumulative effect of different structure irregularities was demonstrated to exhibit non-monotonous dependence on the V layer thickness and achieve a minimal value of 0.31 nm when the thickness of the V layers was 1.2 nm. The discontinuous growth of very thin V films increased in roughness as the thickness of V layers decreased. The columnar growth of the polycrystalline grains in both materials became more pronounced with increasing thickness, resulting in a continuous increase in the interface width to a maximum of 0.9 nm for a 4 nm thickness of the V layer.

## 1. Introduction

The wavelength region of a “water window” (2.3 < λ < 4.4 nm) attracted considerable interest over the last few decades. Extraordinary contrast between the absorptivity of carbon and oxygen provides great opportunities for using soft X-ray microscopy to visualize nanostructures in an aqueous environment inside a cell [1,2,3,4]. Various light sources that can produce coherent soft X-ray in the “water window” region have also been developed [5,6]. Normal incidence optics with multilayer reflecting coating helps to realize good spatial resolution. Eurther, an essential part of the radiation power emitted by a laboratory point source of soft X-rays can be collected. The former is a prime necessity to obtain the images of biological objects in vivo within an extremely short duration [7].

However, a short radiation wavelength (λ) requires the deposition of multilayer structures within an extremely short duration, such that the thickness of an individual layer (~ λ/4) is typically less than 1 nm. Therefore, various irregularities, even on a sub-nanometre scale, in the multilayer structure (e.g., interdiffusion between neighboring ultrathin layers, interfacial roughness, crystallization of layers, possible formation of islands at the initial stages of a film growth, etc.) have a drastic effect on reflectivity. Therefore, experimental normal incidence reflectivity is typically extremely low when as compared with theoretical predictions for an ideal multilayer mirror [8].

Among multilayer mirrors suitable for operating in the “water window” including those of Sc-, Ti-, and V-based, and chromium (Cr)/Sc structure are most extensively studied thus far [9,10,11,12]. The near normal incidence reflectance of 32% was obtained experimentally at the λ = 3.11 nm wavelength [10]. For shorter wavelength intervals of the “water window”, λ = 2.43–2.7 nm, the Cr/V multilayer demonstrates high theoretical reflectance exceeding 60%, while only 9% reflectance at near normal incidence has been achieved in an experiment conducted by [10].

The development of highly reflective short-period Cr/V multilayer mirrors is necessary for the progression of microscopy in the “water window” [13]. Previous works mainly focused on the optical performance of Cr/V multilayers and ways to improve the interface quality of the multilayer with a fixed thickness. To further develop this short period multilayer mirror, essential efforts in the comprehensive study of the growth mechanism of ultrathin Cr and V layers with the layer thickness from sub-nanometer to a few nanometers, and their internal structure and interaction with each other are required. To this end, in the present study, we analyze the microstructure of Cr/V multilayers and its variation as the vanadium (V) layer thickness varies in the range 0.6–4 nm, aiming to determine the optimal conditions of multilayer structure growth. This work will serve as a guide for the further experimental optimization of the multilayer structure, including the usage of different interface engineering methods like inserting diffusion barrier layers, nitridation of the layers, etc. [12]. 

## 2. Experimental Details

The periodic Cr/V multilayer structures were fabricated using a direct current magnetron sputtering technique. Magnetron sputtering is the most widely used technique to fabricate extreme ultraviolet and X-ray coatings around the world. It can provide higher kinetic energy of the deposited atoms comparing to evaporation, which enables a smooth and continuous growth of such thin layers. The magnetron sputtering is a very stable process, so that the deposited film thickness can be controlled only by deposition time. This allows the fabrication of very large multilayer mirrors up to 1 m length or 0.5 in diameter. Comparing to ion beam sputtering, the system is simpler and more cost-effective. It is also characterized by a low temperature growth condition comparing to the chemical vapor deposition methods. Relatively low vacuum due to the injection of sputtering gas can result in the introduction of a small amount of impurities into the multilayer structure. However, this effect is typically negligible in the X-ray region excluding the case of a multilayer operating near the absorption edges of the impurity atoms. The sputtering machine is made in China and was designed by us. The vacuum system includes a molecular pump and a mechanical pump. The base pressure before deposition was 2 × 10^−4^ Pa. High purity argon (99.999%) was supplied as the sputtering gas at a working pressure of 0.1 Pa. The gas flow rate was 20 SCCM. The multilayers were deposited onto super polished silicon wafers with a surface roughness of 0.2 nm (RMS (Root Mean Square)) determined from 1 × 1 μm^2^ AFM (Atomic Force Microscope) scan. The size of the samples was 20 × 20 mm^2^. Switching from Cr to V deposition was provided by a substrate movement in a circular orbit crossing the sputtered area from one magnetron to the other successively. The sputtering power of the two materials is around 20 W. The deposition rate was determined in the preliminary experiment by measuring the film thickness with X-ray reflectometry and divided by the deposition time. The deposition rates of Cr and V were 0.05 and 0.02 nm/s, respectively. The fabricated Cr/V multilayers comprise 30 bilayers with Cr layer placed on top. The V layer thickness varies in the range 0.6–4.0 nm for different samples, while the Cr layer thickness was the same (1.3 nm) for all samples. 

The multilayers were first characterized by the grazing-incidence X-ray reflectometry (GIXR). The measurements were conducted using a laboratory X-ray diffractometer at the Cu-Kα characteristic radiation line (λ = 0.154 nm). The reflectivity curves were fitted using Bede Refs software to determine the layer thickness and interface width [14,15]. The interface width is generally introduced to characterize a cumulative effect of both short scale interfacial roughness and smooth variation of the dielectric constant near the interface arising due to implantation of atoms during deposition and interdiffusion of neighbouring layers. The interface width is inserted into the Nevot–Croce factor to describe the decrease in the amplitude reflectivity of each interface [16], which is represented by σ in this paper. Nevertheless, due to the limitation of the fitting software, it is difficult to determine the Cr-on-V and V-on-Cr interface width separately. Thus, only the average interface width (half of the sum of the two interface widths) is discussed which is more proper to compare the layer quality. 

The surface morphology of multilayers was analysed using atomic force microscopy (AFM) using the instrument of Icon Dimension instrument from Bruker. The layer structure and elemental composition were further characterized by transmission electron microscopy (TEM). Eurther, selected area electron diffraction (SAED) and energy dispersive X-ray spectroscopy (EDX) were conducted simultaneously with the TEM measurements. The TEM samples were prepared by focusing an ion beam using the FEI Helios system. The measurements were conducted using the FEI Tecnai G2 F20 instrument from Materials Analysis Technology Inc. 

## 3. Results and Discussion

A schematic of the layer structure of the fabricated Cr/V multilayer is shown in Figure 1 which includes Cr and V layers and the interface area. The measured reflectivity of multilayers with different V layers thickness are shown in Figure 2 versus the grazing incidence angle (circles). The experimental data were fitted (solid curves) with the use of a two layer model assuming pure Cr and V layers in each period. The diffusion and roughness at interfaces were taken into account by the interface width factor in the model, considering the fact that V can form alloy with Cr and exhibits a large solubility in binary Cr systems [17]. In the case of the sample with ultrathin V layers of the 0.6nm thickness, the average interface width was fitted to be σ_aver._ = 0.49 nm. The relatively wide interfaces can be initiated by the island growth of ultrathin V films at the initial stage resulting in development of short-scale roughness [18,19]. After increasing vanadium layers thickness up to 1.2 nm, the average interface width is reduced down to σ_aver._ = 0.31 nm. This indicates a changeover to a smooth growth of continuous vanadium layers. However, further increase in the thickness of V layers up to 4 nm results in dramatic increase of the average interface width up to σ_aver._ = 0.89 nm demonstrating an essential development of irregularities in multilayer structure.

Figure 3 demonstrates the variation of the interface width with an increasing thickness of the V layers. The average interface width shown in the figure is merely a half of the sum of the interfaces on Cr-on-V and V-on-Cr interfaces. By increasing the V layers thickness d_V_ from 0.6 nm to 1.2 nm, the average interface width decreases and achieves a minimal value (about 0.3 nm) at d_V_ = 1.2 nm. Further increase in V layers thickness results in monotonous enlargement of the average interface width, achieving 0.9 nm at d_V_ = 4 nm.

The surface morphology of Cr/V multilayers with different thickness of V layers is shown in Figure 4. The measurements were conducted using AFM over an 1x1 μm^2^ surface area. The surface of the multilayer sample with the 1.2 nm V layers thickness is relatively smooth with the RMS roughness of only 0.36 nm (Figure 4a). The fine features of the 20–30 nm lateral size and the 2.7 nm peak-to-valley value (PV) are observed on it. The surface morphology is evidently varied (Figure 4b) for the sample with the 4 nm V layers thickness. The lateral size and PV of the surface features are increased to a maximum of 30–40 nm and 4 nm, respectively. The RMS roughness is also increased to a maximum of 0.55 nm. The significant roughening of the sample surface with thicker V layers means that the increased interface width shown in Figure 3 is partially caused by the effect of the roughness enhancement during multilayer structure growth. The two-dimensional power spectral density (PSD) functions of the two samples shown in Figure 4c demonstrate an essential increase in roughness in the range of the spatial frequencies f < 30 μm^−1^ for the sample with the 4 nm V layers thickness. 

The TEM images of the cross sections of Cr/V multilayers with V layers thickness d_V_ = 1.2 nm and d_V_ = 4.0 nm are shown in Figure 5 (bright field images) and 6 (dark field images). The bright field and dark field images were measured by collecting the transmitted beam or diffracted beam, respectively. Thus, in the dark field images, only the crystalline structure that diffracts the electron beam will be highlighted while the amorphous area will remain in dark. Bright columns on the images in Figure 6 give an indication of significant layers of crystallization, which is far more pronounced for the structure with d_V_ = 4.0 nm. 

The internal structure of the multilayer with the V layers thickness d_V_ = 1.2 nm is relatively uniform. Although a few crystalline grains grow across the layer boundaries, most of the grains are still limited within a single layer (Figure 6a). The nanograins of 5–10 nm typical lateral size are observed in several layers. This fact is consistent with the small interface width of the sample (Figure 3) and rather smooth surface of multilayer (Figure 4a).

In contrast, the crystallization degree is much higher for the samples with d_V_ = 4.0 nm (Figure 6b). The columnar growth of the grains of 20–40 nm typical lateral size is evident from the bottom to the top of the multilayers, resulting in features of the same size appearing on the multilayer surface (Figure 4b). Some layer boundaries can still be seen in the image. The layer boundaries are flat in the bottom of the multilayer while they become curved in some local areas in the top.

The SAED (Selected Area Electron Diffraction) images (Figure 6c,d) shows a number of diffraction spots implying a texture of the polycrystalline grains and the fact that both Cr and V layers are partially crystallized even in the sample with 1.2 nm V layers thickness. With the increased thickness of V, the orientations of the crystallization of both Cr and V changed partially. Notice that crystallization of ultrathin V layers was also observed in thin Ni/V multilayers [20]. Although thorough TEM study is essential to observe the defects, it is difficult to see the difference between V and Cr in TEM images due to the weak contrast difference. Various methods are used to characterize this nano-structure of Cr/V multilayers.

The atomic distribution of Cr and V in the multilayer with d_V_ = 4.0 nm is characterized with EDX mapping (Figure 7). The Cr and V layers are displayed in the figure in blue and red, respectively. The figures show that the multilayer structure demonstrates almost perfect growth with abrupt and flat interfaces, however, within the first several bilayers. After growth of 5–6 bilayers, interfaces are deformed, indicating the appearance of bumps (indicated by arrows), which are likely caused by crystalline grain growth in the preferred orientation [21]. Such a bumpy microstructure is progressing with multilayer structure growth and results in significant distortion of interfaces and increasing roughness. This is consistent with the layer boundaries observed in Figure 6b.

Therefore, according to the AFM and TEM results, the main reason of Cr/V multilayer structure deterioration is a columnar growth of polycrystalline grains. The effect is mostly pronounced for multilayers with thick vanadium layers, while Cr/V structure with the 1.2 nm thick vanadium layers also demonstrates a tendency toward columnar growth. The situation is dramatized by the fact that normal incidence mirror should consist of several hundred bilayers to reflect effectively in the “water window” rather than 30 bilayers as in our preliminary study of internal multilayer structure. Therefore, crystallization should be suppressed to fabricate highly reflective Cr/V multilayer. We can think about several possible ways to do it, such as the use of barrier layers [12], or passivation of layers by injection of nitrogen or, perhaps, oxygen into deposition chamber.

## 4. Conclusions

A set of Cr/V multilayer samples with the same thickness (1.3 nm) of Cr layers, while different V layer thickness, varying in the range 0.6–4.0 nm was fabricated and studied using GIXR, AFM, TEM, SAED, and EDX techniques. As the V layers thickness increases from 0.6 nm to 1.2 nm, the multilayers exhibited a relatively smooth growth with the average interface width of 0.31 nm. Further increase in the V layer thickness to a maximum of 4 nm results in the monotonous enlargement of the average interface width to a maximum of 0.9 nm. The main reason of multilayer structure deterioration was determined to be columnar growth of the crystalline grains forming inside the layers of both materials. Therefore, to develop highly reflective Cr/V multilayers for “water window” spectral range, the crystallization needs to be suppressed. The results of this work serve as a guide to fabricate other metal/metal multilayer systems that require smooth interfaces. 

## Figures and Tables

**Figure 1 materials-12-02936-f001:**
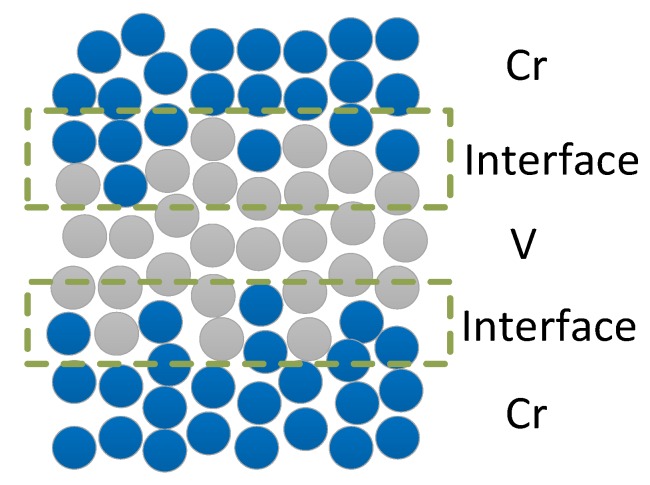
Schematic of the layer structure of Cr/V multilayer.

**Figure 2 materials-12-02936-f002:**
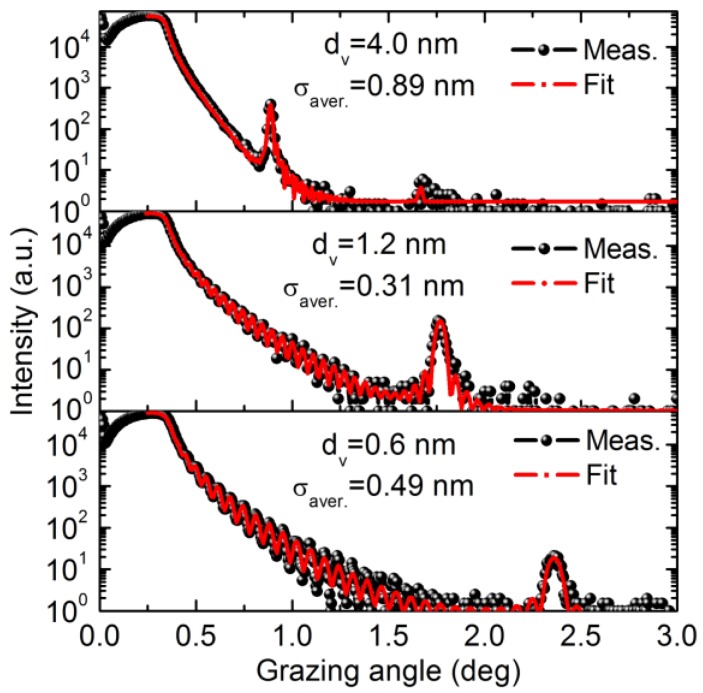
The experimental reflectivity (circles) versus the grazing incidence angle of Cr/V multilayers with the same (1.3 nm) thickness of Cr layers, while varying (from 0.6 nm up to 4 nm) thickness of V layers. The solid curves are the results of fitting.

**Figure 3 materials-12-02936-f003:**
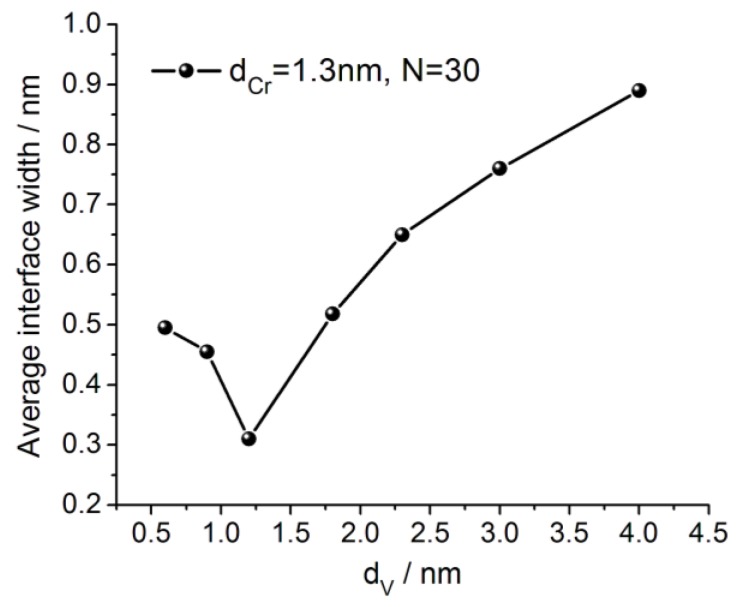
The average interface width versus the vanadium layer thickness.

**Figure 4 materials-12-02936-f004:**
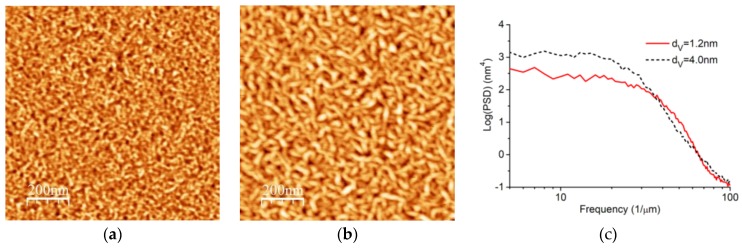
AFM images of the surface of Cr/V multilayers with different vanadium layer thickness: (**a**) d_V_ = 1.2 nm and (**b**) d_V_ = 4.0 nm. The PSD (Power Spectral Density) functions of the surface morphology of the two samples are shown in (**c**).

**Figure 5 materials-12-02936-f005:**
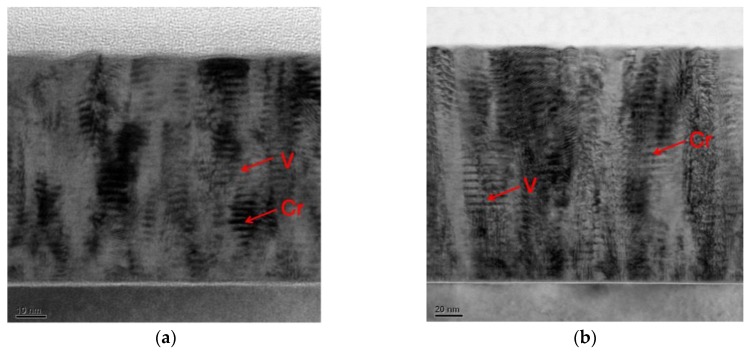
Bright field TEM images of Cr/V multilayer samples with vanadium layers thickness (**a**) d_V_=1.2 nm and (**b**) d_V_=4.0 nm.

**Figure 6 materials-12-02936-f006:**
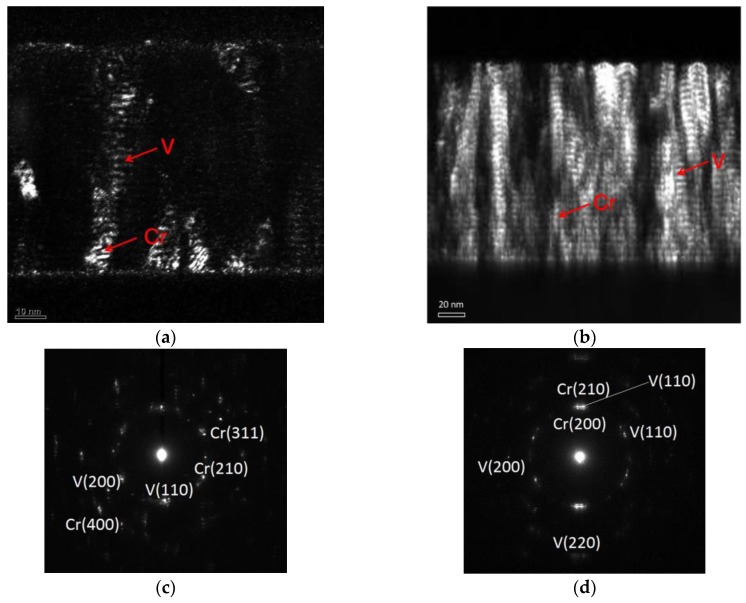
Dark field TEM images of Cr/V multilayer samples with vanadium layers thickness (**a**) d_V_ = 1.2 nm and (**b**) d_V_ = 4.0 nm, and SAED patterns of the same samples with (**c**) d_V_ = 1.2 nm and (**d**) d_V_ = 4.0 nm.

**Figure 7 materials-12-02936-f007:**
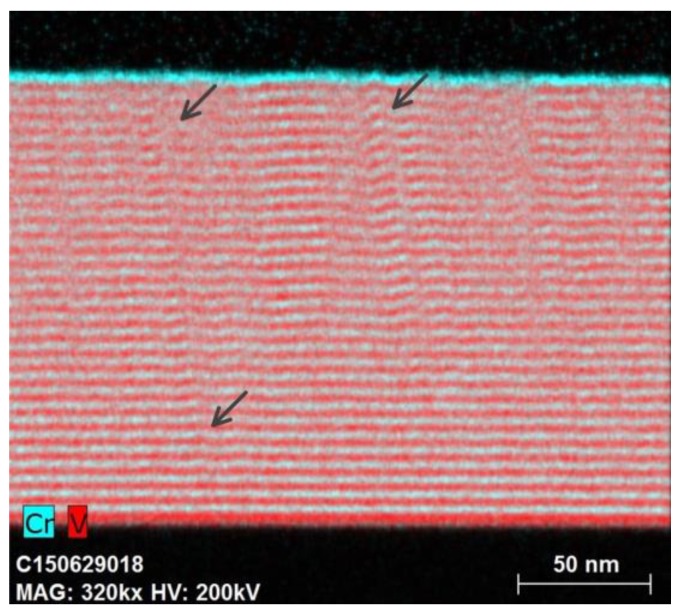
EDX (Energy Dispersive X-Ray Spectroscopy) map of cross section of Cr/V multilayer with d_V_ = 4.0 nm and d_Cr_ = 1.3 nm, where Cr layers are shown in blue and V layers in red. Arrows indicate bumps appeared during multilayer structure growth.

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
