# Peer review of "Evolution of the Internal Structure of Short-Period Cr/V Multilayers with Different Vanadium Layers Thicknesses"

_materials, 2019, doi:10.3390/ma12182936_

Round 1

Reviewer 1 Report

The paper presents new results for deposition of multilayer films suitable for application in x-ray spectroscopy. The paper has many serious flaws which make it unsuitable for publication.

Major points:

Figures are of poor quality.

English needs major revision, either by a professional service and/or a native speaker.

Minor points:

Experimental section: Provide more details of deposition process, e.g., how many magnetrons, magnetron power, how did you do the multilayers (switching from Cr to V). What was the gas flow rate? How did you measure the deposition rate? What is the size of the samples?

Why did you choose magnetron sputtering, i.e., what are the advantages/disadvantages compared to other deposition methods?

Result, first paragraph: It would be helpful to provide a figure showing (a few) layers + interfaces.

Line 81: why a two/layer model if there are 4 layers including two (Cr-on-V and V-on-Cr) interface regions?

Line 94 and figure 1: what is dv, how does it differ from d(V) in figure 1. What are sigmaV and sigmaCr?

Figure 4: what do we learn from this figure?

Line 122 and figures 5 and 6: what is the image size? How can you present such poor quality figures?

In connection with figure 7: I guess, figures 5 and 6 are “top” views while figure 7 is “side” view. Correct? If so, it should be mentioned and also how the sample was prepared for “side” view.

Reviewer 2 Report

The presented manuscript describes the influence of the vanadium thickness on the microstructure of  Cr/V multilayers suitable for applications in the "water window" spectral range. Before publication the following issues should be improved:  

The experimental section should be completed by a short description of device used (model and manufacturer). 2. Fig.6 c,d – the resolution is to low; the indexes are not visible

Reviewer 3 Report

The authors have described the relationship between the thickness and themicrostructure of Cr/V multilayers. This work is interesting, however, it still has lack in experimental measurement prior to the application "water window". Based on literature, there are already Cr/V multilayer approach for "water window" but the authors should highlight their work what the novelty is.

Comments:

The introduction is short and The author should include some important references concerning "water window" application (Gibson et al. Science).  Line 36: Please explain what various irregularities are during fabrication of mulitlayer. Only using ref 6 is not suitable. Line 58: Please explain how the authors will optimize the multilayer structure using different interface engineering methods.   Section 2: Please introduce small details concerning magnetron sputtering device, it is important that the readers are able to redo the authors' experiments. Which chemicals, which vacuum pump, ... Line 76-77: Which type of focused-ion-beam do the authors have use for preparation TEM lamella. What is voltage operation of TEM ? Line 82: Please explain the phase diagram of binary alloys as it is still unclear to the readers. Lots of typo in the text, please recheck this. Line 84: Please define the sigma symbol, it is not explained in the text. Figures are too small. Please use correct size. Please use one form of abbreviation (d(V) vs dv vs dV) in whole text Line 122: The authors have introduced BF and DF images, is it via TEM or STEM technique? Why do the authors introduce these techniques?  Line 125: Please indicate "nano-grains" and "lateral size" in Figure 5-6. Line 132: Please indicate "layer boundaries" in Figure 6. High resolution TEM images are necessary to interpret the observations. Line 135: How do the authors indicate the crystallization degree is higher for the sample with dv=4.0 nm based on these TEM images?  SAED patterns on Figure 6 is not clear enough as the resolution is very poor.  Figure 7: The authors indicate that a bumpy microstructure is observed and results in significant distortion of interfaces. Can the authors prove that via high resolution TEM images?

It should be interesting to the readers that the authors show that their multilayer structure leads to improving the properties prior to "water window" application.

Round 2

Reviewer 1 Report

The paper is improved and now ready for publication.

Author Response

Thank you so much for the careful review of our paper,  your professional suggestions make our work much better than before! Once again, thanks for your work!

Reviewer 3 Report

The authors have improved the manuscript on the basis of comments. However, TEM results are not convinced enough. Due to the weak absorptions, it is difficult to see the difference between V and Cr, but have the authors consider to perform scanning transmission electron microscope as it can show the difference between V and Cr due to Z-contrast. The extra measurements are not necessary but the authors should highlight this that thorough TEM study is essential to observe the defects.

Some minor comments:

Figure 5 and 6: It is difficult to the reader to interpret these images. The authors should mark some info with arrows on these TEM images.

Figure 6: the scale bar of SAED is missing and TEM diffraction patterns are not to be indexed.

Figre 7: Which V and Cr signals were introduced in this work?
